# Magnetically-dressed CrSBr exciton-polaritons in ultrastrong coupling regime

Tingting Wang[1,2,8], Dingyang Zhang[1,8], Shiqi Yang[1,3,8], Zhongchong Lin[1], Quan Chen[4], Jinbo Yang [1], Qihuang Gong [1,5,6], Zuxin Chen[4] ✉, Yu Ye [1,2,5,6] ✉ & Wenjing Liu[1,5,7] ✉

Over the past few decades, exciton-polaritons have attracted substantial research interest due to their half-light-half-matter bosonic nature. Coupling exciton-polaritons with magnetic orders grants access to rich many-body phenomena, but has been limited by the availability of material systems that exhibit simultaneous exciton resonances and magnetic ordering. Here we report magnetically-dressed microcavity exciton-polaritons in the van der Waals antiferromagnetic (AFM) semiconductor CrSBr coupled to a Tamm plasmon microcavity. Using angle-resolved spectroscopy, we reveal an exceptionally high exciton-photon coupling strength, up to 169 meV, demonstrating ultrastrong coupling that persists up to room temperature. By performing temperature-dependent spectroscopy, we show the magnetic nature of the exciton-polaritons in CrSBr microcavity as the magnetic order changes from AFM to paramagnetic. By applying an out-of-plane magnetic field, we achieve effective tuning of the polariton energy while maintaining the ultrastrong exciton-photon coupling strength. We attribute this to the spin canting process that modulates the interlayer exciton interaction.

The strong coupling between photons and matter excitations such as excitons, phonons, and magnons is of central importance in the study of light-matter interactions[1–5]. Bridging the flying and stationary quantum states, the strong light-matter coupling enables the coherent transmission, storage, and processing of quantum information, which is essential for building photonic quantum networks[6,7]. Exciton-polaritons, which combine the high coherence and small effective mass of photons with the large nonlinearity of excitons, have aroused extensive research interest[8,9] and exhibit various micro- and macroscopic quantum phenomena, including photon blockade[10–12], Bose-Einstein condensate[13,14], and superfluidity[15,16]. In particular, the ultrastrong coupling regime is established if the photon-exciton coupling strength is comparable to the frequency of the uncoupled system[7,17], which

could enable the preparation of exotic many-body quantum states with potential applications in quantum computing and simulations. Further dressing the exciton-polaritons with magnetic orders is highly desired in both fundamental and practical aspects, as it can not only induce the magnetic tuning knobs crucial to topological photonics and nonreciprocal photonic devices[18–21] but can also directly access magnonics and spintronics by optical means[22–24]. Moreover, the coherent coupling among distinct excitations, i.e., polaritons and magnons, allows interconnection between multiple frequency ranges, which may establish new paradigms for quantum transducers[25–27]. Previous studies have shown that coupling with cavity photon affects the properties of two-dimensional (2D) magnetic materials (such as linear dichroism response[28]), and numerically predicted that a

[1]State Key Laboratory for Mesoscopic Physics and Frontiers Science Center for Nano-optoelectronics, School of Physics, Peking University, Beijing 100871, China. [2]Collaborative Innovation Center of Quantum Matter, Beijing 100871, China. [3]Academy for Advanced Interdisciplinary Studies, Peking University, Beijing 100871, China. [4]School of Semiconductor Science and Technology, South China Normal University, Foshan 528225, China. [5]Yangtze Delta Institute of Optoelectronics, Peking University, Nantong 226010, China. [6]Liaoning Academy of Materials, Shenyang 110167, China. [7]Collaborative Innovation Center of Extreme Optics, Shanxi University, Taiyuan 030006, China. [8]These authors contributed equally: Tingting Wang, Dingyang Zhang, Shiqi Yang. ✉e-mail: chenzuxin@m.scnu.edu.cn; ye_yu@pku.edu.cn; wenjingl@pku.edu.cn

new polariton mode will appear in the hybrid system composed of topological insulators and 2D magnetic materials[29]. So far, the only reported strong light-matter coupling to a magnetic semiconductor is 2D NiPS$_3$[30], however, no spin-correlation phenomenon has been demonstrated since flipping such an in-plane zigzag antiferromagnetic (AFM) order[31,32] requires a high magnetic field up to ~6 T[33], while a much higher magnetic field (~10.5 T) is required for optical probing[34,35].

CrSBr is a 2D van der Waals (vdW) material that combines strong excitonic physics and layered A-type AFM order[36,37]. Arranged into a layered orthorhombic crystal stacked along the $c$ axis, each layer of CrSBr is ferromagnetically ordered along with the easy axis ($b$ axis) and antiferromagnetically coupled to the adjacent layers[36]. CrSBr is a direct bandgap semiconductor with a highly anisotropic band structure, and the 1s excitons corresponding to its energy band Γ point are linearly polarized along the $b$ axis, exhibiting a large exciton binding energy of about 0.5 eV for a monolayer while 0.25 eV for bulk crystal[37,38]. Pioneering studies have revealed that CrSBr excitons are strongly coupled to its magnetic order through spin-alignment-dependent interlayer interactions[38-41].

In this work, we demonstrate magnetically dressed exciton-polaritons in CrSBr thin flakes. An ultrastrong coupling regime is reached in cavity-coupled CrSBr flakes with a giant coupling strength of up to 169 meV, owing to the tightly confined vdW excitons. We show that these polaritons can be effectively engineered by controlling the magnetic order of CrSBr on the parameter space of temperature and magnetic field.

## Results and Discussion

Bulk CrSBr exhibits an excitonic complex dominated by the 1s exciton state around 1.37 eV at cryogenic temperatures, accompanied by several unidentified excitonic resonances at slightly higher energies[42]. Knowledge of the refractive index and extinction coefficient of CrSBr is essential before the study of exciton-polariton but has not been reported. Here we measured the angle-resolved reflectance spectra of CrSBr flakes transferred onto silicon substrates at 7 K, and fitted the refractive index using the transfer matrix method (TMM). Considering the excitonic nature of CrSBr, Lorentzian dispersion is implemented to describe the material as follows

$$\varepsilon = \varepsilon_{\mathrm{bg}} + \sum_j \frac{f_j/\hbar^2}{\omega_j^2 - \omega^2 - i\Gamma_j\omega},$$ (1)

where $\varepsilon$ denotes the material permittivity, $\varepsilon_{\mathrm{bg}}$ is the background permittivity that accounts for the optical responses other than the

excitons, $\omega_j$, $\Gamma_j$, and $f_j$ are the angular frequency, damping rate, and the oscillator strength of the $j^{th}$ excitonic state, respectively. Typical reflectance spectra and their corresponding fitting curves for linear polarization along the $a$ and $b$ axes are presented in Fig. 1b. The reflectance spectrum polarized along the $a$ axis is featureless and can be fitted solely by a background permittivity $\varepsilon_{\mathrm{bg}} = 10.0$, given the thickness of the CrSBr flake measured to be 177 nm by atomic force microscope. In contrast, the reflectance spectrum along the $b$ axis becomes highly oscillating, and the model fitting resembles the experimental data by taking into account two excitonic states, identified as the 1s (X) and X* excitons at 1.366 eV and 1.381 eV, respectively. X excitons are found to possess a large oscillator strength of $f_X = 1.7\,(\mathrm{eV})^2$ and narrow linewidth of $\Gamma_X = 0.68$ meV, enabled by the tight bounding nature of excitons in vdW materials. X* excitons have smaller oscillator strength of $f_{X^*} = 0.5\,(\mathrm{eV})^2$ and larger linewidth of $\Gamma_{X^*} = 7.5$ meV. The background permittivity along the $b$ axis is calculated as $\varepsilon_{\mathrm{bg}} = 11.1$. The extracted parameters are confirmed in several flakes of different thicknesses (from 200 nm to 440 nm), with typical oscillator strengths ($f_{\mathrm{ex}}$) of X exciton in the range of $1.7-2.1\,(\mathrm{eV})^2$ (see Supplementary Information Fig. S1).

Strong coupling experiments are then carried out on Tamm plasmon microcavities, where CrSBr flakes are placed on distributed Bragg reflector (DBR) substrates and covered with 45 nm silver thin films (Fig. 1a). The specific CrSBr flake under investigation has a uniform thickness of 462 nm (Fig. 2a) across both lateral dimensions over 20 μm (Fig. 2a) and thus can be treated as a 2D system. Exciton-photon coupling is studied by angle-resolved reflectance and photoluminescence (PL) spectroscopy measurements at 7 K, as presented in Fig. 2b and c, respectively. Up to 7 mode branches can be clearly resolved, with rapidly decreasing mode spacing as they asymptotically approach the excitonic energy at 1.370 eV. Such asymptotic behavior avoids crossing between the photonic and excitonic modes, which is one of the significant characteristics of strong exciton-photon coupling. Besides, the mode branches near the exciton state are significantly flattened, manifesting that light-matter hybridization increases the effective mass as the modes become more exciton-like.

To emphasize the large anisotropy of the CrSBr excitons, the linear polarization-dependent reflectance spectroscopy is performed on the coupled structure (a polarizer is inserted in the collection light path), as shown in Fig. 2d. With the polarization along the $a$ axis of the CrSBr flake, one reflection dip at 1.408 eV is observed, assigned to the photonic cavity mode with a quality factor around 350, corresponding to $\Gamma_{\mathrm{ph}} = 2.0$ meV. When the polarization is swept towards the $b$ axis of the CrSBr flake, polariton mode series emerge, further confirming that

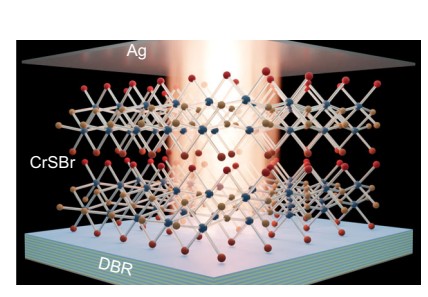

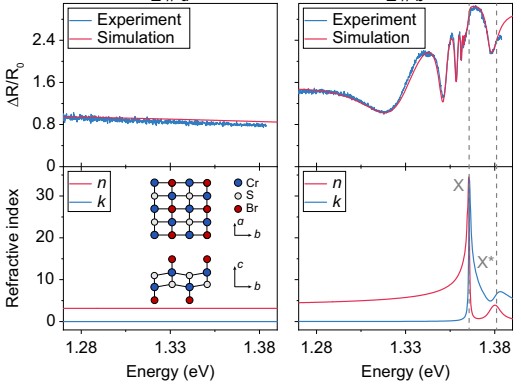

**Fig. 1 | Device structure and optical characterizations of CrSBr. a** Schematic of a CrSBr flake coupled to a Tamm plasmon microcavity, composed of a bottom DBR, a thin CrSBr flake, and a top silver mirror. **b** Upper panel: experimental and fitted reflectance spectra of a CrSBr flake with a thickness of 177 nm along the $a$ (left) and $b$ (right) axes at normal incidence. The reference reflection $R_0$ is taken on the bare Si substrate. Lower panel: calculated refractive index ($n$) and extinction coefficient ($k$) of CrSBr along the $a$ (left) and $b$ (right, containing two excitons) axes. Inset: crystallographic structure of CrSBr.

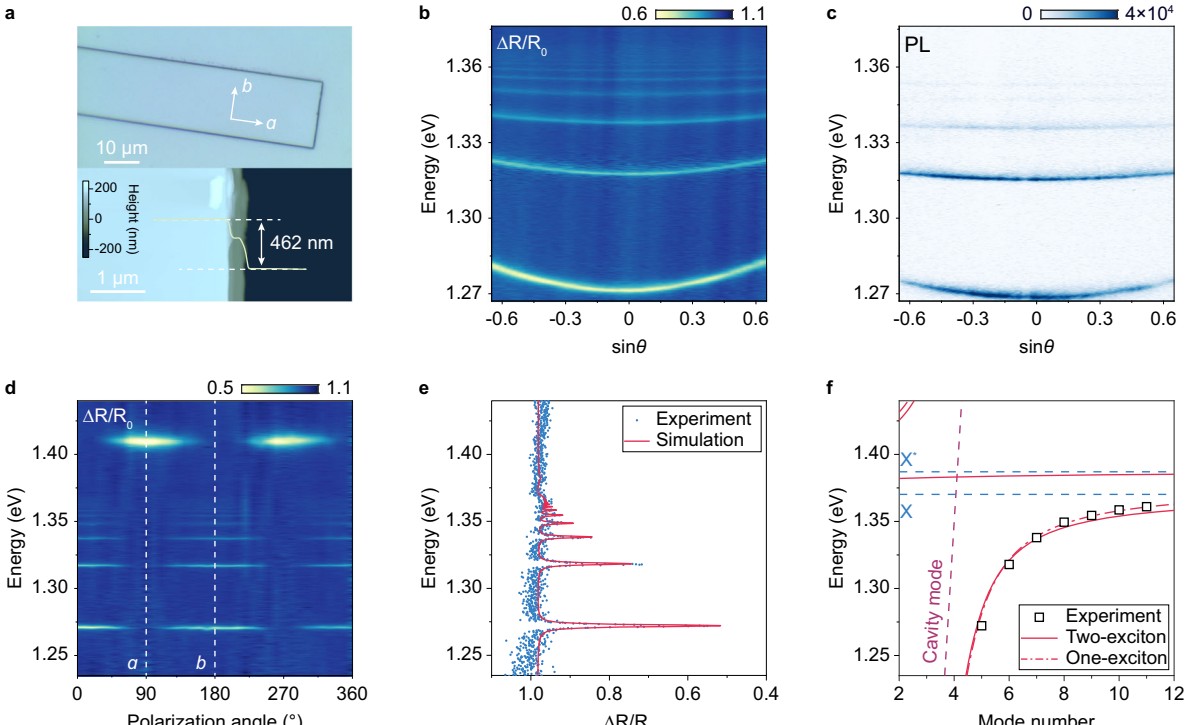

**Fig. 2 | Ultrastrong exciton-polariton coupling in cavity-coupled CrSBr.**
**a** Optical microscope image of the CrSBr microcavity (upper panel) and an atomic force microscope image of the CrSBr flake in the microcavity (lower panel). **b** Angle-resolved reflectance spectrum of the cavity-coupled CrSBr flake. **c** Angle-resolved PL spectrum of the cavity-coupled CrSBr flake. **d** Polarization-dependent reflectance spectra at normal incidence, illustrating the large anisotropy of CrSBr excitons. **e** Reflectance spectrum polarized along the *b* axis at normal incidence and its corresponding TMM fitting results. **f** Mode number-dependent exciton-polariton energy and COM fitting to polariton dispersion utilizing two-exciton and one-exciton Lorentzian models.

they originate from the coupling to CrSBr excitons. The stark contrast between the photonic and polaritonic dispersion reflects the giant exciton oscillator strengths and extremely strong exciton-photon coupling in CrSBr.

Quantitatively, TMM analysis is performed to extract the exciton oscillator strength via fitting the experimental reflectance spectrum, as presented in Fig. 2e. The excitonic refractive index is described by Eq. (1) that simultaneously incorporates the X and X* excitons. The calculations yield excellent fits to the experiments, and in this specific sample, the exciton oscillator strengths and linewidths are extracted to be $f_X = 2.0$ (eV)$^2$, $f_{X^*} = 0.6$ (eV)$^2$, $\Gamma_X = 0.6$ meV, and $\Gamma_{X^*} = 8.5$ meV. The complete angle-resolved spectra are calculated by finite-difference time-domain (FDTD) simulations, which demonstrate good agreement with the experimental results, as shown in Supplementary Information Fig. S2. Furthermore, to extract the exciton-photon coupling strength, a coupled oscillator model (COM) is utilized, including the coupling between both the X and X* excitons and the photonic modes, as described by the Hamiltonian

$$H = \hbar \sum_i \omega_i a_i^\dagger a_i + \hbar \omega_{ph}(\mathbf{k}) b^\dagger b + \sum_i g_i(a_i^\dagger b + b^\dagger a_i). \quad (2)$$

Here, $a_i^\dagger$ and $b^\dagger$ denote the creation operators of the $i^{th}$ excitons and cavity photons, **k** is the in-plane wave vector of the cavity photons, $\omega_i$ and $\omega_{ph}$ are their respective angular frequencies, and $g_i$ is the exciton-photon coupling strength. In the calculation, the exciton energies and polariton mode orders are obtained from the TMM and FDTD simulations. The dispersion of the photonic cavity modes is calculated from the microcavity with the same geometry parameters while replacing the excitonic material by a dielectric layer with $\varepsilon = \varepsilon_{bg}$, as explained in detail in Methods and Supplementary Information Fig. S3. The fitting results are presented in Fig. 2f, where all the

observed polariton modes are identified and well-fitted by COM. Due to the complexity of the energy spectrum above the X and X* excitons in CrSBr, the upper polariton branches have not been resolved. A giant coupling strength of $g_X = 169$ meV is found for the X exciton-photon coupling, and $g_{X^*}$ is calculated to be $g_{X^*} = 92$ meV, in accordance with their respective oscillator strengths. These large coupling strength values not only ensure that the system is in the strong coupling limit of $g \gg (\Gamma_X, \Gamma_{ph})$, but the large ratio of $g > 0.1\omega_X$ also indicates that the coupled system enters the ultrastrong coupling regime, where exotic quantum optical phenomena can be expected. Moreover, as the coupling strength $g$ is much larger than the detuning between the X and X* excitons of 15 meV, the polariton dispersion can be approximately described by the one-exciton Lorentzian model, as plotted by the red dash-dotted line in Fig. 2f. The one-exciton model yields a coupling strength of $g = 193$ meV, which matches the summation of the oscillator strengths of the X and X* excitons. Therefore, this coupling strength can represent the overall coupling between cavity photons and the CrSBr exciton complex with effective energy at $\omega_{ex}$ and will be used in the following discussions.

To understand the effect of magneto-electronic coupling in CrSBr on the exciton-polaritons, we first performed temperature-dependent reflectance measurements. The magnetic ordering of CrSBr is characterized by a Néel temperature ($T_N$) of 132 K, above which it transitions from AFM to paramagnetic[36,38], as experimentally verified by the temperature-dependent magnetic susceptibility measurements presented in Supplementary Information Fig. S4. Across the phase transition, a change in the exciton peak shape has been reported[38] due to the strong magneto-electronic coupling in CrSBr, and thus the modulation of the polariton properties can be anticipated. The cavity-coupled sample under investigation has a similar thickness of 460 nm, with both lateral dimensions exceeding 60 μm (see Supplementary Information Fig. S5). The temperature-dependent

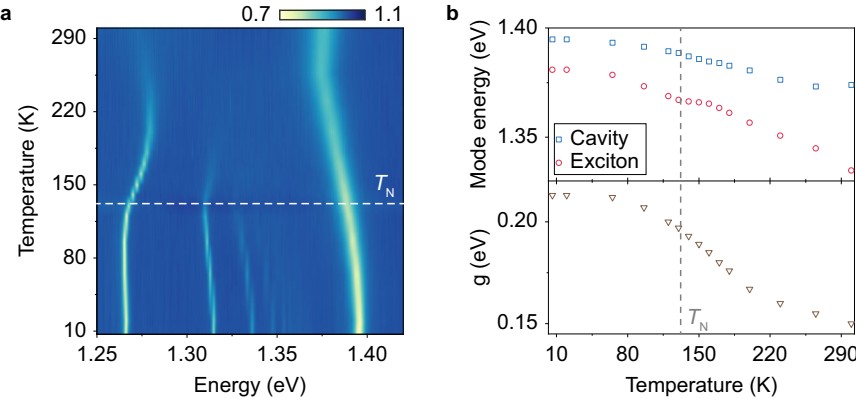

**Fig. 3 | Exciton-polariton evolution with temperature. a** Unpolarized reflectance spectrum of a cavity-coupled CrSBr flake at normal incidence as a function of temperature. **b** The extracted temperature-dependent evolution of the uncoupled cavity mode, the excitonic mode, and the exciton-photon coupling strength.

unpolarized reflectance spectra at normal incidence are summarized in Fig. 3a. At 7 K, the mode at 1.39 eV is identified as the uncoupled photonic cavity mode polarized along the $a$ axis of CrSBr. As the temperature increases from 7 K to room temperature, it redshifts monotonically to 1.37 eV owing to thermal expansion and changes in permittivity. In contrast, for the polariton branches located below 1.37 eV at 7 K, while they also redshift at low temperatures, a turning point appears around $T_N$, followed by a prominent blueshift from 130 K to 220 K, above which the redshift recovers till room temperature (Fig. 3a). This kink near $T_N$ suggests that the properties of the exciton-polaritons are tailored by the magnetic order of the material.

COM fitting is applied to the angle-resolved reflectance spectrum at each temperature to gain further insight into the temperature-dependent polariton evolution. Assuming that the variation of the background permittivity $\varepsilon_{bg}$ is consistent along different polarizations, the bare photonic component can then be calculated from the uncoupled cavity mode along the $a$ axis. The exciton energy $\omega_{ex}$ and coupling strength $g$ are obtained from the fitting. As shown in Fig. 3b, the calculated $\omega_{ex}$ redshifts with increasing temperature and exhibits a kink at $T_N$ due to the change of interlayer excitonic coupling, in accordance with previous reports[38]. Meanwhile, the coupling strength remains nearly constant up to 70 K, and then decreases monotonically with increasing temperature. Therefore, near the magnetic phase transition point, the near-constant exciton energy and the rapid drop in the coupling strength result in the blueshift of the polariton branches. Above 220 K, the redshifts of both the excitonic state and the polariton modes recover. TMM simulation is utilized to calculate the exciton energy and oscillator strength where consistent results are obtained. Remarkably, the obtained large coupling strength of about 150 meV maintains up to room temperature, thereby highlighting the giant exciton binding energy and ultrastrong light-matter coupling in CrSBr. The same measurements have been conducted on another CrSBr flake (620 nm) placed on SiO₂/Si substrate without an external microcavity (see Supplementary Information Fig. S6). Interestingly, the thick CrSBr flake itself can serve as an intrinsic microcavity that supports dielectric leaky modes, which in turn leads to the emergence of exciton-polaritons (see Supplementary Information Fig. S6a). To highlight the role of the Tamm plasmon cavity, we applied FDTD simulations to compare the performance in them of coupling strength, mode quality factor, field distribution, etc., with and without the cavity (see Supplementary Information Fig. S6b-d and Fig. S7). The external cavity is found to strongly enhance both the temporal and spatial confinements of the photonic modes, leading to more efficient exciton-photon strong coupling.

Finally, the magnetic tuning of the exciton-polaritons is characterized at various temperatures below and above $T_N$. Applying a

magnetic field along the $c$ axis, it is reported that the magnetic order of CrSBr experiences a transition from the A-type AFM to the forced ferromagnetic (FM) phase through the continuous spin-canting process. To understand the magnetic tuning effect of exciton-polaritons in the cavity-coupled system, Fig. 4a–c present the reflectance spectra at normal incidence as a function of the applied magnetic field. At $T = 2$ K, all the polariton branches undergo continuous redshift with the increase of the applied magnetic field until they are saturated at around 2.4 T. A similar trend is found at $T = 100$ K, but with a lower saturation magnetic field of 1.1 T. Contrarily, at $T = 150$ K above $T_N$, the polariton branches only slightly redshift with the applied magnetic field without saturation.

The magnetic tuning of the polariton branches perfectly matches the evolution of the magnetic order of the CrSBr flake, as confirmed by the reflective magnetic circular dichroism (RMCD) spectroscopy. Figure 4d–f show the RMCD signals of a CrSBr flake with similar thickness under a magnetic field applied along the $c$ axis at the three selected temperatures. To minimize the effects caused by the strong optical birefringence in CrSBr, we made the polarization of the probe beam parallel to the crystallographic axis of the sample ($a$ axis or $b$ axis) to obtain the RMCD signal. As a typical in-plane A-type AFM, CrSBr exhibits null net magnetic moment at $\mu_0 H = 0$ T, followed by a spin-canting process along the hard axis under an external field along the $c$ axis. At 2 K, the saturation field $\mu_0 H_{sat}$ is identified at 2.4 T, as shown by the dashed gray lines in Fig. 4d, which is consistent with the saturation field exhibited by the exciton-polaritons. Both $\mu_0 H_{sat}$ and the polariton saturation field decrease with increasing temperature and reach 1.1 T at 100 K. Finally, when the temperature reaches 150 K (above $T_N$), the RMCD signal evolves linearly with the magnetic field, corresponding to the paramagnetic state. The same evolution trends of the RMCD signal and the exciton-polaritons illustrate that the polaritonic properties are directly correlated to the magnetic order of the CrSBr and can be effectively tuned via the applied magnetic field. Figure 4g–i show the calculated exciton energies and coupling strengths for different temperatures. The energy shifts in the polariton branches with magnetic field are determined to arise from the exciton energy shift, which is caused by the enhanced interlayer excitonic coupling induced by spin canting[38]. To be specific, the interlayer coupling is described as $\langle S_1 | S_2 \rangle \propto \cos(\theta/2)$, where $\theta$ denotes the angle between magnetization vectors of adjacent layers. Thus, a shift in energy as a quadratic function of the magnetic field is expected, as confirmed by the calculated exciton energies at 2 K and 100 K (Fig. 4g, h). This result is in accordance with the previous report by Wilson et al. on bare CrSBr flakes[38]. On the other hand, the coupling strength $g$ remains nearly constant in the AFM and FM phases of CrSBr, confirming the robustness of the strong exciton-photon coupling under magnetic tuning.

In conclusion, we demonstrate ultrastrongly coupled exciton-polaritons in CrSBr that can be effectively tailored by the interlayer magnetic order. These findings open up the possibilities to rich many-body physics in hybrid quantum systems involving the coherent interactions between distinct quasiparticles across a wide range of energies. The exciton-mediated coupling of photonic architectures to the magnetic orders may restore hindered magnetic responses in passive photonic systems in the optical wavelength range. This could enable photonic devices with new functionalities, such as non-reciprocal transmissions, nontrivial topologies, and magnetically tunable Bose-Einstein condensates. vdW CrSBr can be further integrated with other 2D materials into 2D heterostructures that allow highly integrated on-chip opto-electronic-magnetic devices.

## Methods

### Sample preparations and characterizations

**Crystal growth.** CrSBr single crystals were grown using the chemical vapor transport method. Disulfur dibromide and chromium metal were mixed in a molar ratio of 7:13, then sealed in a silica tube under vacuum. Thereafter, the evacuated silica tube was placed in a two-zone tubular furnace. CrSBr crystals were grown under a temperature gradient of 950 °C to 880 °C for 7 days, with a heating/cooling rate of 1 °C/min.

**Microcavity preparations.** The bottom DBR substrates are composed of 9 pairs of 160 nm $SiO_2$/100 nm $TiO_2$ thin films. To obtain large and flat CrSBr flakes, the CrSBr bulk was exfoliated onto the polydimethylsiloxane (PDMS) films with the help of blue and white tapes. Then, the PDMS substrates with CrSBr flakes adhered were transferred onto the target substrates, e.g., DBR, Si or $SiO_2$/Si substrates, via a dry-transfer method. Finally, a 45 nm silver film was evaporated using electron beam evaporation with a deposition rate of 1 Å/s.

**Atomic force microscope characterization.** The thicknesses of CrSBr flakes were characterized by an atomic force microscope (Cypher, Asylum Research) with tapping mode.

### Optical measurements

**Angle-resolved reflectance and PL measurements.** Reflectance and PL measurements were carried out in Montana cryostat (Montana CR-563) with a base temperature of 7 K. For reflectance measurements, a broadband white light source was focused onto the sample with an objective (N.A. = 0.65). For PL measurements, the sample was excited by a pulsed laser (5 MHz, 100 ps) centered at 650 nm. The same objective was used to collect the reflected light/PL signal and the back-focal plane was projected onto the entrance slit of a spectrometer (SR-500i) equipped with 300 lines $mm^{-1}$ grating, and the angle-resolved spectra were collected by an electron multiplying charge-coupled device (EMCCD, Andor Ultra 888).

**Magnetic filed-dependent measurements.** Magnetic field-dependent measurements were carried out based on the Attocube closed-cycle cryostat (attoDRY 2100) with a base temperature of 1.6 K

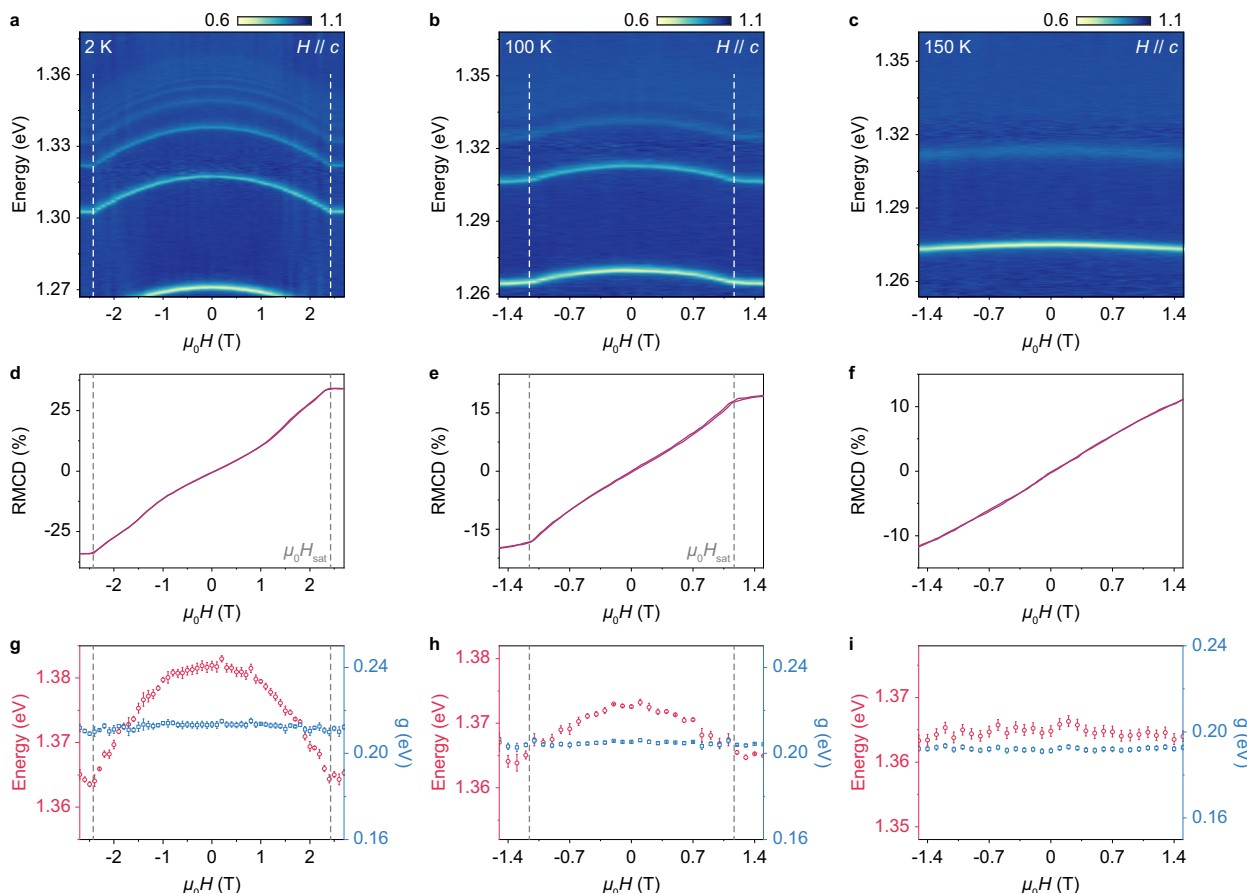

**Fig. 4 | Magnetic field control of CrSBr exciton-polaritons. a–c** The magnetic field-dependent (along the *c* axis) polariton reflectance spectra at normal incidence recorded at 2 K (**a**), 100 K (**b**), and 150 K (**c**), respectively. **d–f** RMCD signals of a CrSBr flake of similar thickness placed on $SiO_2$/Si substrate as a function of the applied magnetic field measured at 2 K (**d**), 100 K (**e**), and 150 K (**f**), respectively. **g–i** Exciton energy and coupling strength *g* values from COM fittings using the one-exciton Lorentzian model, corresponding to the results in **a–c**.

and a maximum out-of-plane magnetic field of 9 T. The cavity-coupled sample was kept in this cryostat equipped with an objective (N.A. = 0.82) for magnetic field-dependent angle-resolved reflectance measurements. The spectra were measured by an Andor spectrometer (SR-500i-D2-R) equipped with a grating of 600 lines mm⁻¹ and a Newton CCD. For the RMCD measurements, a linearly polarized 633 nm HeNe laser was modulated between left and right circular polarization by a photoelastic modulator (PEM) at 50.052 kHz and a chopper at a frequency of 795 Hz. The modulated light was focused by the objective onto the sample, and the reflected signals were separated from the incident part by a beam splitter and detected by a photomultiplier tube. The magnetization information of the sample was detected by the RMCD signal due to the polar magneto-optic effect, which was determined by the ratio of the a.c. component from the PEM at 50.052 kHz and the a.c. component from the chopper at 795 Hz. These signals were measured by a two-channel lock-in amplifier (Zurich HF2LI).

**Data analysis**

**TMM.** The TMM was applied to calculate the reflectance from the multilayer thin film structure of cavity-coupled CrSBr. Specifically, at normal incidence, the transfer matrix of the $j^{th}$ layer with thickness $d_j$ and complex refractive index $N_j = n_j - ik_j$ at wavelength $\lambda$ is written as

$$M_j = \begin{pmatrix} \cos \delta_j & \frac{i \sin \delta_j}{N_j} \\ iN_j \sin \delta_j & \cos \delta_j \end{pmatrix}, \qquad (3)$$

where $\delta_j = \frac{2\pi}{\lambda} N_j d_j$. The transfer matrix of the system containing multiple layers can be expressed as

$$M = \prod_j M_j = \begin{pmatrix} M_{11} & M_{12} \\ M_{21} & M_{22} \end{pmatrix}. \qquad (4)$$

The reflection coefficient is hence

$$r = \frac{N_0 M_{11} + N_0 N_s M_{12} - M_{21} - N_s M_{22}}{N_0 M_{11} + N_0 N_s M_{12} + M_{21} + N_s M_{22}}, \qquad (5)$$

in which $N_0$ and $N_s$ are the complex refractive index of the vacuum and the substrate, respectively.

**FDTD simulation.** FDTD simulations were performed by implementing the refractive index of the two-exciton Lorentzian model. The simulated cavity-coupled CrSBr structure was assumed to be infinitely large in the in-plane dimensions, which was excited by an angle-varying plane-wave source. A far-field power monitor was implemented to collect the reflected signal.

**Fitting to the coupled oscillator model.** The bare photonic cavity modes were obtained by calculating the Tamm plasmon mode dispersions while replacing the CrSBr flake with a dielectric layer with $\varepsilon = \varepsilon_{bg} = 11$, as shown in Supplementary Information Fig. S3. Cavity modes were calculated via the TMM while sweeping the thickness of the dielectric CrSBr layer. According to the electric field distribution, the modes were assigned to different mode families with the mode numbers $m$ labeled in Supplementary Information Fig. S3b. Each mode family was fitted to a linear dispersion relation, from which the photonic mode energies associated with the CrSBr thickness were calculated. According to the linear fitting, an empirical formula of the dispersion relation can be written as

$$m\lambda_c = 2n_{bg}(d + \delta d), \qquad (6)$$

where $\lambda_c$ is the wavelength of the cavity mode, $n_{bg}$ is the background refractive index of the dielectric CrSBr, $m$ is the mode number, $d$ is the thickness of the dielectric CrSBr, $\delta d$ is an effective cavity length induced by the electric field penetrating into the DBR structures. Specific to the DBR substrate used in our study, $\delta d = 95$ nm, which is nearly independent of $m$. For the exciton energies, in the two-exciton model, the energies of the X and X* excitons were extracted from the TMM fitting, and thus the coupling strength $g$ was the only fitting parameter in the COM. In the one-exciton model, both $\omega_{ex}$ and $g$ were fitted from the COM. The fittings were performed according to Eq. (2) via the Problem-based Optimization Workflow provided by Matlab utilizing the least square method.

## Data availability
The source data generated in this study have been deposited in the Zenodo database under accession code https://doi.org/10.5281/zenodo.8137336. Source data are provided with this paper.

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

## Acknowledgements

This work was supported by the National Key R&D Program of China (Grant No. 2022YFA1206700, and 2018YFA0306900), the National Natural Science Foundation of China (Grants Nos. 12241401, 11975035, 61875001, 12250007, 62175002, 62035017, and 92150301), the China Postdoctoral Science Foundation (2023TQ0003), and the Beijing Natural Science Foundation (Grant No. JQ21018).

## Author contributions

T.W., W.L. and Y.Y. conceived the project, designed the experiments, and analyzed the results. T.W. and W.L. conducted the reflectance and PL measurements. T.W., S.Y. and W.L. conducted the magneto-optic measurements. S.Y. conducted the RMCD measurement. D.Z. and W.L. performed the theoretical models and calculations. Q.C. and Z.C. grew the CrSBr bulk crystals. Z.L. and J.Y. performed the magnetic characterizations of the bulk crystals. T.W., S.Y., W.L. and Y.Y. drafted the manuscript. All authors discussed the results and contributed to the manuscript.

## Competing interests

The authors declare no competing interests.
