## [Peer Review File · Nature Communications]

Reviewers' Comments:

Reviewer #1:

Remarks to the Author:

The authors have addressed most of the concerns raised by the prior reviewers. Some of the literature references and discussion with coupling of photons with magnetic 2D materials still remains somewhat incomplete (e.g. : *Nature Photonics* 16 (4), 311-317, 2022; *Phys. Rev. Materials* 7, 045201, 2023). After addressing this, I think this manuscript is suitable enough for publication in *Nature Communications*.

Reviewer #2:

Remarks to the Author:

I appreciate the authors' responses to my comments. I have a few questions and comments as follows:

1. The authors' main claim is ultrastrong coupling in the CrSBr system. Can you compare the coupling strength quantitatively with other systems?
2. In the extended data Figure 6, the bare CrSBr flakes without a cavity already seem to have multiple branches, which might be from exciton-polariton states. In this case, the role of the Tamm plasmon cavity is not very clear. Do you have any comments on this? Could you compare the coupling strength with and without the cavity?
3. The authors presented the AFM data to support the claim of flatness for the flakes. However, as shown in extended data Figure 1, the reflectance depends sensitively on thickness. There could be multiple steps inside the flakes that can affect the data. The current AFM image is not clear enough to confirm the flatness of the measured point. I suggest the authors consider changing the scale bar or presenting an error bar to provide further support for their claim.

Responses to Reviewer #1

Comments: *The authors have addressed most of the concerns raised by the prior reviewers. Some of the literature references and discussion with coupling of photons with magnetic 2D materials still remains somewhat incomplete (e.g. Nature Photonics 16 (4), 311-317, 2022; Phys. Rev. Materials 7, 045201, 2023). After addressing this, I think this manuscript is suitable enough for publication in Nature Communications.*

Responses: We thank the reviewer for his/her positive evaluation and are glad that the reviewer found our work to be “*suitable enough for publication in Nature Communications*” after adding an appropriate discussion of the literature. We thank the reviewer for his/her valuable suggestions to cite these representative and pioneering studies we missed. Following the reviewer’s suggestions, we have carefully read these works and discussed them in the main text. Nature Photonics 16, 311-317, (2022) and Phys. Rev. Materials 7, 045201, (2023) are cited as new references 28 and 29, respectively. These changes are marked in the revised manuscript.

Responses to Reviewer #2

General comments: *I appreciate the authors' responses to my comments. I have a few questions and comments as follows.*

Responses: We thank the reviewer for his/her constructive comments and are glad that the reviewer appreciated our responses to his/her comments. Below we provide detailed point-by-point responses to the reviewer's comments.

Comments: *The authors' main claim is ultrastrong coupling in the CrSBr system. Can you compare the coupling strength quantitatively with other systems?*

Responses: We thank the reviewer for his/her constructive comments. To address the reviewer's concerns, we summarize the exciton-photon coupling strengths in various exciton-polariton systems as Table 1. Comparing the exciton-photon coupling strength in various microcavity exciton-polariton systems, it can be seen that the ultra-strong coupling strength we achieved in the CrSBr microcavity is higher than that in most inorganic material systems. The organic excitonic systems with strongly confined Frankel excitons can support higher coupling strengths, but usually at the expense of larger exciton linewidths.

In addition to the ultra-strong coupling strength, the exciton-polaritons in CrSBr can be effectively tailored by the interlayer magnetic order. By applying an out-of-plane magnetic field, an efficient tuning of the polariton energy (energy tuning range of 31.5 meV with a linewidth of 1.9 meV) is achieved, which is attributed to the spin canting process that modulates the interlayer exciton interaction. Endowing new degrees of freedom to the light-matter coupling may propose a new paradigm to modify the nature of light and matter.

Comments: *In the extended data Figure 6, the bare CrSBr flakes without a cavity already seem to have multiple branches, which might be from exciton-polariton states. In this case, the role of the Tamm plasmon cavity is not very clear. Do you have any comments on this? Could you compare the coupling strength with and without the cavity?*

Responses: We thank the reviewer for the constructive comments. As the reviewer mentioned, the thick CrSBr flake itself can indeed serve as an intrinsic microcavity that supports dielectric leaky modes, which in turn leads to the emergence of exciton-polaritons. Nevertheless, a properly designed external cavity can significantly enhance the performance of exciton-polaritons by improving both the temporal and spatial confinements. We applied finite-difference

Table 1: Comparison of exciton-photon coupling strengths in various microcavity exciton-polariton systems. QW: quantum well. RT: room temperature, 300 K. MW: microwire. NW: nanowire. ML: monolayer. NP: nanoplate.

Material	Morphology	Temperature (K)	g (meV)	Reference
GaAs	QWs	4	7.5	[1]
GaAs	QWs	20	8.7	[2]
GaAs	QWs	77	9.5	[3]
CdTe	QWs	5	13	[4]
GaN	QWs	RT	28	[5]
GaN	MW	RT	9.5	[6]
ZnO	Bulk	RT	49	[7]
ZnO	Bulk	130	65	[8]
ZnO	NW	RT	50	[9]
MoS ₂	ML	RT	23	[10]
WS ₂	ML	RT	50	[11]
WS ₂	Flake	RT	116	[12]
CsPbBr ₃	NP	RT	60	[13]
CsPbBr ₃	NW	77	100	[14]
CsPbCl ₃	NP	RT	132.5	[15]
MAPbBr ₃	NW	RT	195	[16]
MAPbBr ₃	Film	RT	35	[17]
(PEA) ₂ PbI ₄	Flake	4	55	[18]
(PEA) ₂ PbI ₄	Flake	RT	65	[19]
J aggregates	Film	RT	200	[20]
J aggregates	Film	RT	80	[21]
TDAF	Film	RT	495	[22]
CrSBr	Flake	7	169	this work

time-domain (FDTD) simulations to compare the quality factor and field distribution of the photonic modes of the Tamm plasmon cavity and the SiO₂/Si substrate, as shown in Fig. R1. For both cases, the thickness and refractive index of the CrSBr flake (without exciton) is taken to be 460 nm and $\epsilon_{bg} = 11$, respectively, as in the main text of the manuscript. The simulated reflectance spectra of the two configurations are shown in Fig. R1, with quality factors of 410 and 12 extracted for the Tamm cavity and no cavity cases. Furthermore, the spatial distribution of the electric field of their photonic modes is calculated and plotted, as

shown in Fig. R1b. With a Tamm cavity, the electric field is mostly confined in the CrSBr flake, whereas without a cavity, the mode leaks significantly into the SiO₂ layer and into the air. These differences suggest that the external cavity can enhance the coupling strength and lifetime of the polaritons through better spatial and temporal confinements, which may be advantageous in polariton lasing, condensation, and other optoelectronic applications.

Figure R1: Comparison of photonic modes with and without the Tamm plasmon cavity. **a.** Simulated reflectance spectra of a 460 nm dielectric layer sandwiched in a Tamm plasmon cavity (top) and placed on a SiO₂/Si substrate (bottom). Photonic modes are identified as dips in the reflectance spectra. **b.** Electric field amplitude distributions for the Tamm plasmon mode (top) and dielectric leaky mode (bottom), where the dielectric leaky mode is marked by the vertical dashed line in the bottom panel of **a**. In the calculations, the permittivity of the dielectric layer is taken as the background permittivity of CrSBr $\epsilon_{bg} = 11.0$.

Following the reviewer's suggestion, we calculated the coupling strengths of the sample without the external cavity in Extended Data Fig. 6 (Supplementary Information Fig. S6), as shown in Fig. R2. The fitting procedure is the same as that in Fig. 2 in the main text, where a two-exciton model is used for FDTD simulations ($\epsilon_{bg} = 11.0$, $f_X = 2.37$ (eV)², $f_{X^*} = 0.5$ (eV)², $\Gamma_X = 1.0$ meV, and $\Gamma_{X^*} = 5.0$ meV) and compared with the experimental reflectance spectrum, as shown in Fig. R2a to confirm a good match. The mode number of each mode was then identified by the calculated spatial field distribution. The corresponding photonic mode dispersion was simulated by replacing CrSBr with a dielectric

with $\epsilon_{bg} = 11.0$, same as in Fig. R1a. The fitted coupling strength is $g_X = 170$ meV, which is almost the same as 169 meV of the sample in Fig. 2 of the main text, despite the fact that the CrSBr flake here is much thicker (620 nm) and of larger exciton oscillator strength ($f_X = 2.37$ (eV)²) comparing to the one used in Fig. 2 (460 nm thick, $f_X = 2.00$ (eV)²). To highlight the role of the Tamm plasmon cavity, we also calculated the coupling strength of a hypothetical CrSBr flake on the SiO₂/Si substrate, with the same thickness and material parameters as those in Fig. 2, and obtained a smaller coupling strength of $g_X = 154$ meV.

These corresponding discussions and Figs. R1 and R2 have now been added to the revised supplementary information.

Figure R2: Characterization of exciton-photon coupling in the sample in Extended data Fig. 6 (Supplementary Information Fig. S6). **a.** Reflectance spectrum polarized along the b axis at normal incidence and its corresponding FDTD simulation results. **b.** Mode number-dependent exciton-polariton energy and polariton dispersion fitted using the coupled oscillator model two-exciton Lorentzian model.

Comments: *The authors presented the AFM data to support the claim of flatness for the flakes. However, as shown in extended data Figure 1, the reflectance depends sensitively on thickness. There could be multiple steps inside the flakes that can affect the data. The current AFM image is not clear enough to confirm the flatness of the measured point. I suggest the authors consider changing the scale bar or presenting an error bar to provide further support for their claim.*

Responses: We thank the reviewer for his/her constructive comments. As suggested by the reviewer, we changed the scale bars of the atomic force microscope images shown in Fig. 2a and Supplementary Information Fig. S5, so that the flatness of the CrSBr flakes can be clearly illustrated. In addition, we also analyzed the surface roughness of the samples. In an area of $\sim 2 \times 2 \mu\text{m}^2$, the surface roughness (expressed by the arithmetic mean deviation, R_a) of the CrSBr flake shown in Fig. 2a is $R_a \approx 1.61 \text{ nm}$ ($R_a \approx 1.44 \text{ nm}$ for the sample in Supplementary Information Fig. S5), which is negligible relative to the sample thickness. Moreover, no multiple steps were observed inside the CrSBr samples.

References

- [1] H. Deng, G. Weihs, C. Santori, J. Bloch, and Y. Yamamoto, *Science* **298**, 199–202 (2002).
- [2] S. Brodbeck, S. De Liberato, M. Amthor, M. Klaas, M. Kamp, L. Worschech, C. Schneider, and S. Höfling, *Physical Review Letters* **119**, 027401 (2017).
- [3] J. Bloch, T. Freixanet, J. Marzin, V. Thierry-Mieg, and R. Planel, *Applied Physics Letters* **73**, 1694–1696 (1998).
- [4] J. Kasprzak, M. Richard, S. Kundermann, A. Baas, P. Jeambrun, J. M. J. Keeling, F. Marchetti, M. Szymańska, R. André, J. Staehli, et al., *Nature* **443**, 409–414 (2006).
- [5] G. Christmann, R. Butté, E. Feltin, J.-F. Carlin, and N. Grandjean, *Applied Physics Letters* **93**, 051102 (2008).
- [6] A. Trichet, F. Médard, J. Zúñiga-Pérez, B. Alloing, and M. Richard, *New Journal of Physics* **14**, 073004 (2012).
- [7] T.-C. Lu, Y.-Y. Lai, Y.-P. Lan, S.-W. Huang, J.-R. Chen, Y.-C. Wu, W.-F. Hsieh, and H. Deng, *Optics Express* **20**, 5530–5537 (2012).
- [8] T. Guillet, M. Mexis, J. Levrat, G. Rossbach, C. Brimont, T. Bretagnon, B. Gil, R. Butté, N. Grandjean, L. Orosz, et al., *Applied Physics Letters* **99**, 161104 (2011).

- [9] L. K. van Vugt, S. Rühle, P. Ravindran, H. C. Gerritsen, L. Kuipers, and D. Vanmaekelbergh, *Physical Review Letters* **97**, 147401 (2006).
- [10] X. Liu, T. Galfsky, Z. Sun, F. Xia, E.-c. Lin, Y.-H. Lee, S. Kéna-Cohen, and V. M. Menon, *Nature Photonics* **9**, 30–34 (2015).
- [11] Z. Sun, J. Gu, A. Ghazaryan, Z. Shotan, C. R. Consideine, M. Dollar, B. Chakraborty, X. Liu, P. Ghaemi, S. Kéna-Cohen, et al., *Nature Photonics* **11**, 491–496 (2017).
- [12] H. Zhang, B. Abhiraman, Q. Zhang, J. Miao, K. Jo, S. Roccasecca, M. W. Knight, A. R. Davoyan, and D. Jariwala, *Nature communications* **11**, 3552 (2020).
- [13] R. Su, J. Wang, J. Zhao, J. Xing, W. Zhao, C. Diederichs, T. C. Liew, and Q. Xiong, *Science Advances* **4**, eaau0244 (2018).
- [14] T. J. Evans, A. Schlaus, Y. Fu, X. Zhong, T. L. Atallah, M. S. Spencer, L. E. Brus, S. Jin, and X.-Y. Zhu, *Advanced Optical Materials* **6**, 1700982 (2018).
- [15] R. Su, C. Diederichs, J. Wang, T. C. Liew, J. Zhao, S. Liu, W. Xu, Z. Chen, and Q. Xiong, *Nano Letters* **17**, 3982–3988 (2017).
- [16] S. Zhang, Q. Shang, W. Du, J. Shi, Z. Wu, Y. Mi, J. Chen, F. Liu, Y. Li, M. Liu, et al., *Advanced Optical Materials* **6**, 1701032 (2018).
- [17] P. Bouteyre, H. S. Nguyen, J.-S. Lauret, G. Trippe-Allard, G. Delport, F. Lédée, H. Diab, A. Belarouci, C. Seassal, D. Garrot, et al., *ACS Photonics* **6**, 1804–1811 (2019).
- [18] L. Polimeno, A. Fieramosca, G. Lerario, M. Cinquino, M. De Giorgi, D. Ballarini, F. Todisco, L. Dominici, V. Ardizzone, M. Pugliese, et al., *Advanced Optical Materials* **8**, 2000176 (2020).
- [19] A. Fieramosca, L. Polimeno, V. Ardizzone, L. De Marco, M. Pugliese, V. Maiorano, M. De Giorgi, L. Dominici, G. Gigli, D. Gerace, et al., *Science Advances* **5**, eaav9967 (2019).
- [20] H.-S. Wei, C.-C. Jaing, Y.-T. Chen, C.-C. Lin, C.-W. Cheng, C.-H. Chan, C.-C. Lee, and J.-F. Chang, *Optics Express* **21**, 21365–21373 (2013).

- [21] D. G. Lidzey, D. Bradley, M. Skolnick, T. Virgili, S. Walker, and D. Whittaker, *Nature* **395**, 53–55 (1998).
- [22] S. Kéna-Cohen, S. A. Maier, and D. D. Bradley, *Advanced Optical Materials* **1**, 827–833 (2013).

Reviewers' Comments:

Reviewer #2:

Remarks to the Author:

The authors have appropriately replied to the questions. I think the manuscripts are suitable for publication in Nature Communications.

Responses to Reviewer #2

Comments: *The authors have appropriately replied to the questions. I think the manuscripts are suitable for publication in Nature Communications.*

Responses: We are glad that the reviewer is satisfied with our responses and find our work to be “*suitable for publication in Nature Communications*”. We would like to sincerely thank the reviewer for his/her constructive comments, which are crucial to improving the quality as well as integrity of our work.